# Risk assessment at work and prevention strategies on COVID-19 in Italy

**Sergio Iavicoli, Fabio Boccuni◉\*, Giuliana Buresti, Diana Gagliardi◉,**
**Benedetta Persechino, Antonio Valenti, Bruna Maria Rondinone**

Department of Occupational and Environmental Medicine, Epidemiology and Hygiene, INAIL, Monte Porzio Catone, Rome, Italy

\* f.boccuni@inail.it

**Data Availability Statement:** All relevant data are within the paper and its Supporting Information files.

**Funding:** The authors received no specific funding for this work.

## Abstract

The COVID-19 pandemic has spread worldwide, with considerable public health and socio-economic impacts that are seriously affecting health and safety of workers, as well as their employment stability. Italy was the first of many other western countries to implement extended containment measures. Health workers and others employed in essential sectors have continued their activity, reporting high infection rate with many fatalities. The epidemiological trend highlighted the importance of work as a substantial factor to consider both when implementing strategies aimed at containing the pandemic and shaping the lockdown mitigation strategy required for sustained economic recovery. To support the decision-making process, we have developed a strategy to predict the risk of infection by SARS-CoV-2 in the workplace based on the analysis of the working process and proximity between employees; risk of infection connected to the type of activity; involvement of third parties in the working processes and risk of social aggregation. We applied this approach to outline a risk index for each economic activity sector, with different levels of detail, also considering the impact on mobility of the working population. This method was implemented into the national epidemiological surveillance model in order to estimate the impact of re-activation of specific activities on the reproduction number. It has also been adopted by the national scientific committee set up by the Italian Government for action-oriented policy advice on the COVID-19 emergency in the post lockdown phase. This approach may play a key role for public health if associated with measures for risk mitigation in enterprises through strategies of business process re-engineering. Furthermore, it will make a contribution to reconsidering the organization of work, including also innovation and fostering the integration with the national occupational safety and health (OSH) system.

## Introduction

The COVID-19 pandemic has spread worldwide reporting more than 16 million people infected in over 200 countries at the date of the present work [1], with considerable public health and socio-economic impacts that are also seriously affecting health and safety of workers, as well as their employment stability.

**Competing interests:** The authors have declared that no competing interests exist

In this respect, most countries have adopted containment measures, including social distancing, telework and suspending of diverse non-essential work activities [2].

Italy was the first among western countries to face the spread of the pandemic, and one of the most severely hit at global level. The number of hospital admissions for COVID-19 has considerably challenged the national health system's capacity to respond to patients' needs with particular reference to the availability of intensive care unit beds [3].

The progressive adoption of several containing measures by the Italian Government included the temporary suspension of most of business activities, resulting in a reduction of about 75% of workers present in their workplaces as of March 25th. According to the estimates reported by the National Institute of Statistics (ISTAT), the whole non-suspended sectors comprised 2.3 million companies (51.2% of the total), accounting for 15.6 million workers (66.7% of the total), while suspended employees were about 7.8 million (33.3%) [4].

Schools and universities were closed: teachers and students continued their activities by means of e-learning tools. It has been estimated that approximately 25% of the employees attended their workplaces (e.g. health facilities, security forces, army, food supply chain, pharmacies, transports, etc.) as smart working and annual leave incentives were widely adopted by public administration and many private businesses.

As a result, the epidemiological data showed a low level of infection transmission, which initiated a progressive release of containment measures with a stepwise approach of progressive lifting of the lockdown according to the model suggested by WHO, and guided by the risk management model described in the present paper.

The overall containing measures aimed to guarantee workers' health and safety and to reduce social contacts for the whole population. The measures aimed at also tackling the risk of infection intrinsic to any work activity. Health workers and others employed in essential sectors have continued their activity, despite facing several organizational challenges, including the critical shortage of personal protective equipment [5–7]. The epidemic spreading among health workers brought to light that the risk of infection related to work is very concrete. As confirmed by the latest data available, such situation caused a very high number of infections among health workers equal to 12.2% of the total cases. Several fatalities were also recorded [8]. Such phenomenon is common to other countries hit by the pandemic [9].

In other sectors outbreaks of the virus took place among workers in meat and poultry processing facilities in USA [10] and other countries. Cases of COVID-19 have been observed in other aggregation settings, including correctional and detention facilities [11] and homeless shelters [12]. Tourism, retail and hospitality industry, transport and security workers, and construction workers was recognized as probable occupationally acquired COVID-19 in a Singapore study [13]. In Italy the impact of COVID-19 on workers may be also measured by over 47,000 compensation claims and 208 deaths related to occupational exposure to COVID-19 registered at May 31st [14].

Work-related exposure can occur anytime at the workplace, during work-related travel to an area with local community transmission, as well as on the way to and from the workplace [15]. Epidemiological data show an increased risk of poor outcome by age and comorbidity [8] in the general population, making these individuals more fragile and vulnerable to infection, also in the work context.

These figures highlight the importance of work as a substantial factor to consider both when implementing strategies aimed at containing the pandemic and shaping the lockdown mitigation strategy required for sustained economic recovery.

As the reproduction number (Rt) was below 1 and the pandemic reached a steady state of low-level transmission, the debate focused on how to balance the gradual and controlled lifting of the containment measures, while guaranteeing a continued public health protection policy.

The virus will continue circulating until an effective vaccine will be available or herd immunity will be achieved; thus, there is a persistent risk of new disease outbreaks that will need to be counterbalanced by rigorous interventions, which should involve both public and occupational health.

Guidance documents for workplace safety and health have been published at international level [16]. COVID-19 risk categorization approaches have been proposed in which work-related exposure depends on the probability of coming into close or frequent contact with people who may be infected by SARS-CoV-2 or with contaminated surfaces and objects [15].

In this framework, the present study, describes a stepwise approach, based on a methodology to assess the occupational risk of infection by SARS-CoV-2. As long as proper epidemiological indicators are met, this approach can ensure a safe return to work after the lockdown, guaranteeing specific standards in terms of workers' health and safety.

The major findings of this approach have been adopted by the Italian Government for action-oriented policy in order to determine priority and interventions on the COVID-19 emergency [17].

## Materials and methods

A method to estimate the risk of infection by SARS-CoV-2 in the workplace has been developed taking into account, on one hand, the specific characteristics of production processes and the impact of work organization on the risk; on the other hand, we considered that many jobs require close contact with external subjects (public, clients, etc.), which increases the likelihood of social aggregation, with consequences that may easily expand towards the community.

This methodology is based on the general approach to risk analysis in the occupational safety and health (OSH) field [18]. In this case, such approach is not strictly intended to mitigate harm for single work activities; instead, it is aimed at identifying the general integrated occupational risk levels for the working population by economic sector in line with the strategy of the decision makers for the lifting of the containment measures.

The occupational risk of infection by SARS-CoV-2 has been classified based on three variables:

- **Exposure**: the likelihood to be in contact with potential source of infection during the work activity, according to the scale from 1 = "not exposed" to 5 = "completely exposed".

- **Proximity**: the intrinsic features of work activity which cannot guarantee an adequate social distancing. The parameter was graded according to the scale from 1 = "work carried out alone almost throughout the working time" to "5 = "work carried out in close proximity with others for most of the working time".

- **Aggregation**: the condition linked to work activities that may determine contacts with people other than workmates (restaurants, retail, entertainment, hospitality, education, etc.) defined as a factor in the following classes: 1.00 = "limited presence of a third party" (e.g. manufacturing sector, industry, offices that are not opened to the public); 1.15 = "intrinsic presence of third parties controlled through the organization" (e.g. retail, personal services, offices that are opened to the public, cafes, restaurants); 1.30 = "aggregations controllable with procedures" (e.g. health care, schools, prisons, army, public transports); 1.50 = "large aggregations not easily controllable by specific procedures" (e.g. shows, sport events).

The first two parameters represent respectively the probability of contact with potential sources of infection and the physical proximity to other people during work. For example, a

microbiologist may have a high index of exposure due to his specific activities but lower index of physical proximity to other colleagues; instead, a dancer or an actor may have little probability of encountering potential sources of infection but comes inevitably into close contact with other workers.

To quantify such parameters, we used the proximity and exposure perception indicators defined by the Occupational Information Network (O*NET) online database, based on the Standard Occupational Classification (SOC) and available for over 900 professions [19]. We translated the SOC occupations into the Italian Classification of Economic Activities–ATECO 2007, derived from the European Classification of Economic Activities (NACE Rev.2) [20], through clerical coding methods as described by Mannetje and Kromhout [21] and already used for epidemiological studies.

Both exposure and proximity average values have been calculated for each employment sector according to ATECO classification. The scales were normalized using the following equation:

$$y_i = \frac{(x_i - x_{min})}{(x_{max} - x_{min})} \qquad (1)$$

where $y_i$ is the standardized score for the $i^{th}$ sector, $x_i$ is the original rating score, $x_{min}$ is the lowest possible score on the rating scale used, and $x_{max}$ is the highest possible score on the rating scale.

To evaluate the reliability in the Italian context, we compared the O*NET perception indicator of exposure to the indicator of exposure to biological risk (viruses or bacteria) already defined in the framework of the Italian Survey of Occupational Safety and Health at Work (INSuLa) for each ATECO sector. This survey is based on a representative sample of national working population and it is periodically repeated every 5 years [22, 23]. Similarly, for the physical proximity indicator we applied the comparison with the indicator used in the Italian Sample Survey on Professions (ICP) [24]. In both cases, the Pearson correlation coefficient was statistically significant with values of 0.794 (p<0.001) and 0.625 (p = 0.003) respectively.

The third parameter is the social aggregation connected to the job, rating it from scant presence of a third party (e.g. on an assembly line) to large aggregations not easily controlled by specific procedures (e.g. sport events). The aggregation factor category has been defined for each employment sector based on its characteristics and adapted from the classification of occupant load factors (S1 and S2 Tables in S1 Appendix) for business activities already established by technical regulations at national and international level [25, 26]. The final product defines the risk levels (R) in four classes: Low R < 2; Medium-Low 2 < R < 4; Medium-High 4 < R < 8; High R > 8 within the iso-risk curves, as reported in S1 Fig in S1 Appendix.

Furthermore, updated data on the workforce [4] were associated to each activity sector to obtain a burden of risk levels related to the number of potential exposed workers. The link between the amount of people working in each suspended sector during the national lockdown, with the commuting variables (such as the percentage of use of public transportation by sector and hourly distribution of mobility) allowed us to highlight the potential impact on the mobility due to the reactivation of businesses and commuting. Last analysis was performed by gender, class of age and geographical area.

Based on this risk matrix approach, measures have been identified to prevent/mitigate the risk of infection for the workers and the community at large.

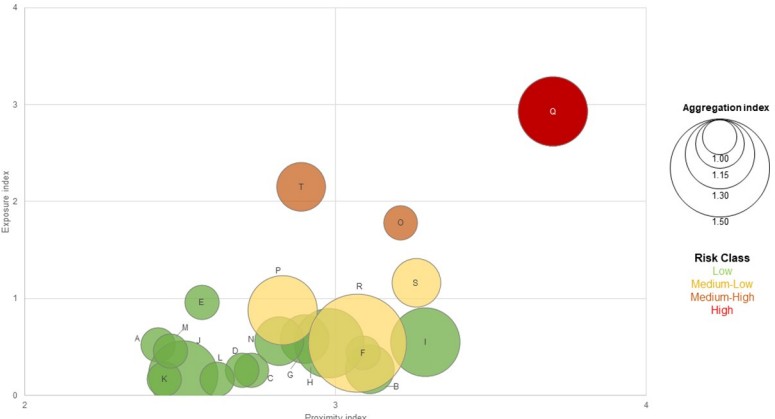

**Fig 1. Risk class per employment sector.** Low R< 2; Medium-Low 2≤R<4; Medium-High 4≤R<8; High R≥8.

## Results

We identified the risk class of each employment sector with the allocation of a color code (Fig 1). The different size of the bubbles is directly proportional to the aggregation factor assigned to each sector.

Table 1 below illustrates the risk classes for the employment sectors according to the first level of the ATECO classification and their partitions, along with the related number of workers.

**Table 1. Risk class and working population by employment sector.**

| | Description of employment sectors (ATECO classification) | Risk class | No. of workers (per 1,000) |
|---|---|---|---|
| A | Agriculture, forestry and fishing | Low | 908.8 |
| B | Mining and quarrying | Low | 24.7 |
| C | Manufacturing | Low | 4,321.4 |
| D | Electricity, gas, steam and air conditioning supply | Low | 114.1 |
| E | Water supply; sewerage, waste management and remediation activities | Low | 242.8 |
| F | Construction | Low | 1,339.4 |
| G | Wholesale and retail trade; repair of motor vehicles and motorcycles | Low | 3,286.5 |
| H | Transportation and storage | Low | 1,142.7 |
| I | Accommodation and food services activities | Low | 1,480.2 |
| J | Information and communication | Low | 618.1 |
| K | Financial and insurance activities | Low | 635.6 |
| L | Real estate activities | Low | 164.0 |
| M | Professional, scientific and technical activities | Low | 1,516.4 |
| N | Administrative and support services activities | Low | 1,027.9 |
| O | Public administration and defense; compulsory social security | Medium-High | 1,242.6 |
| P | Education | Medium-Low | 1,589.4 |
| Q | Human health and social work activities | High | 1,922.3 |
| R | Arts, entertainment and recreation | Medium-Low | 318.2 |
| S | Other services activities | Medium-Low | 711.6 |
| T | Activities of households as employers; undifferentiated goods and services producing activities of households for own use | Medium-High | 738.9 |
| U | Activities of extraterritorial organization and bodies | Low | 14.1 |

According to the proposed risk classification, health and social work activities (employment sector Q) resulted in higher average risk; activities of households and public administration (employment sectors T and O) at medium-high risk; education, arts, entertainment and recreation and other services (employment sectors P, R, S) at medium-low risk while for all further sectors the average risk resulted as low.

Detail figures are also reported in S1 Appendix. The economic sectors and the indication of the dimension of social aggregation and the integrated average risk class have been calculated for the 2-digit ATECO code to further specify the risk class of work activities included in each economic sector (S3 Table in S1 Appendix). Analysis referred to the 3-digit for sector G has been made, in order to provide an example of a broader specific risk classification of the wholesale sector based on the different commercial activities (S4 Table in S1 Appendix). The related distribution of the workers of suspended sectors during the lockdown phases, classified by gender, age, and geographical areas are also allowed (S5 Table in S1 Appendix).

We employed this approach to establish a risk index for each activity, with different levels of detail and in terms of the working population or their mobility. This has been implemented also in epidemiological models to estimate the impact on the Rt of re-activation of specific activities, during the post-lockdown phase.

This model was also adopted by the national scientific committee set up by the Italian Government for action-oriented policy advice on the COVID-19 emergency. The production activities with low or medium-low risk were prioritized in the gradual process of reshaping the containment measures, along with a suitable and shared prevention strategy that was also aimed at controlling related aggregation risks.

Similar evaluations apply to some activities in the trade and services sectors, where the need to evaluate the impact on mobility and to ensure social distancing has been taken into account. The risks related to worker's mobility and commuting required specific interventions in the public transportation services, with the introduction of appropriate prevention measures.

The progressive re-activation of all economic activities were accompanied by the enforcement of specific safety and health guidelines for each sector all based on the proposed methodological approach.

Finally, according to the risk categorization by employment sector, prevention strategies have been proposed to mitigate further the level of risk by introducing accurate preventive strategies to be applied in the post lockdown phase as summarized in Table 2.

## Discussion

To support the decision-making process, we have developed a strategy to predict the risk of infection by SARS-CoV-2 in the workplace, considering that many jobs require close contact with workmates or outside subjects, increasing the likelihood of social aggregation, with consequences that may easily involve the community.

With this in mind, we classified the risk as the result of interaction among three parameters: exposure, proximity and aggregation. Exposure and proximity indexes are calculated on the basis of perception reporting surveys with a certain confidence interval, even if based on representative samples of the national working population. The aggregation factor is what we call the added value in COVID-19 occupational risk analysis, and might assume a different scale and modularity, in relation to the areas where productive sites are based, the kind of work organization and the preventive measures adopted.

The numerical datum that summarizes the risk level, is an average value for each employment sector that does not allow to highlight the specificity of the risk level associated with some specific activities. This can be highlighted only through a subsequent breakdown at the

**Table 2.  Measures for risk mitigation in enterprises and impacts at community level.**

| Measures for risk mitigation in enterprises |
| --- |
| **Administrative measures to manage times and spaces at work** |
| • Promoting alternative ways of delivering work (e.g. telework or smart working). |
| • Changing the allocation of working spaces to guarantee social distancing. |
| • Re-definition of working times and shifts, fostering flexibility. |
| • Implementation of technological innovations, such as those involving connectivity and automation. |
| **Preventive and protective measures** |
| • Engagement and participative approach of OSH players (e.g. safety managers and workers' safety representatives) and strengthening their role in the enterprise. |
| • Specific information and training to enhance workers' awareness. |
| • Promotion of health behaviors, including social distancing, hand hygiene, and possible use of face masks. |
| • Collective and individual protection measures (e.g. sanitation of work environments). |
| **Measures for vulnerable workers** |
| • Active engagement of occupational health physicians in the implementation of health surveillance measures tailored for fragile or vulnerable workers and patients returning to work after recovering from COVID-19. |
| • Drafting of a social security plan to support vulnerable workers excluded from work. |
| **Impacts at community level** |
| **Mitigation of the effects of mobility** |
| • Prevention of social aggregations in public transport during peak commuting hours. |
| • Integration of business re-opening strategies with municipality mobility plans. |
| • Support for alternative means of mobility (e.g. walking, bike or moto-scooter) |
| **Integration with the overall public health strategy** |
| • Prevention of the emergence of new epidemic clusters, by means of: massive temperature monitoring in the workplace, timely laboratory testing strategies, and effective contact tracing, links with community medicine. |

first and second digit of ATECO classification, which can be assigned in order to provide a greater level of detail where required.

The results of risk classification based on the described methodology supported the National Government in identifying the priorities and the modulation of containing measures, as well as the impact that the reactivation of one or more economic sector implied for the population at large. It is understood that among those sectors where contacts with third parties are more frequent, there are some activities that on their own might determine the reactivation of mobility and huge aggregations (e.g. transportation, wholesale, education and recreation sectors).

Epidemiological indicators are essential to guide each containment measures' mitigating step also in the world of work. The epidemic trend, which demonstrated the effect of the containment measures, required a thoughtful analysis based on the modularity of re-activation of production activities. As shown in Fig 2 the stepwise re-opening of selected working sectors did not produce relevant effects on the epidemic trend. A strong contribution to the containment of infection has been given by the provision of widespread use of medical face masks for all workers in all workplaces and of community masks in all enclosed public places (including retail shops, restaurants, public offices, means of transportation) and outdoor when social distancing cannot be guaranteed. Wearing face masks in public has been recognized as the most effective means to prevent interhuman virus transmission with an estimated reduction of at least 78.000 new cases in Italy between April 6 and May 9 [27].

Risk analysis revealed that many of the most hazardous sectors are among those that have remained open because they are essential. During the first phase of the emergency, several organizational measures in terms of prevention and protection were put in place, as necessary

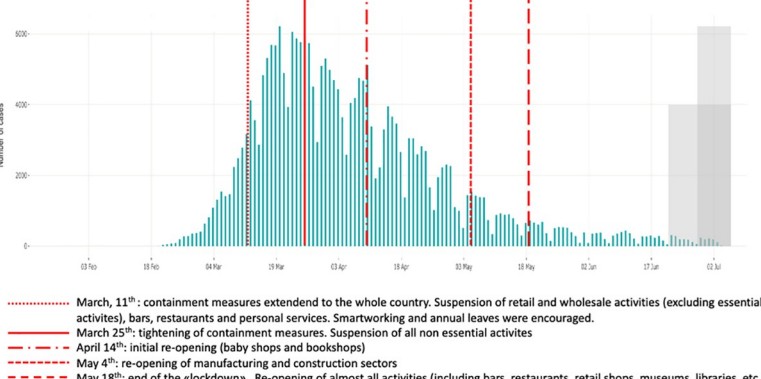

March, 11th : containment measures extendend to the whole country. Suspension of retail and wholesale activities (excluding essential activites), bars, restaurants and personal services. Smartworking and annual leaves were encouraged.
March 25th: tightening of containment measures. Suspension of all non essential activites
April 14th: initial re-opening (baby shops and bookshops)
May 4th: re-opening of manufacturing and construction sectors
May 18th: end of the «lockdown».  Re-opening of almost all activities (including bars, restaurants, retail shops, museums, libraries, etc.)

**Fig 2. Epidemiological trend and major lockdown/re-opening measures.** Adapted from [8]. Note: more recent data (grey square) should be interpreted with caution due to the possible reporting delay of more recently diagnosed cases and to the possibility that cases with data of onset within the reporting period have not yet been diagnosed.

to guarantee safe working conditions for those productive sectors which remained active. As for the health workers, many guidelines were issued by World Health Organization (WHO), European Centre for Disease Prevention and Control (ECDC), European Agency for Safety and Health at Work (EU-OSHA) [2, 16, 28].

In the phase of lifting of containment measures with persistent SARS-CoV-2 circulation, guidelines and procedures were adopted in different scenarios (e.g. retail shops and commercial galleries, restaurants, beaches) in order to mitigate the risk, as defined by this methodology. Basic principles that guided the prevention strategies were: (i) social distancing; (ii) hands, personal and workplace hygiene; (iii) outbreak control capacity. In this framework policy, administrative, hygiene, preventive, protective and communication measure may be implemented to mitigate the risk. The provision of widespread use of medical face masks for all workers in all workplaces and of community masks in all enclosed public places (including retail shops, restaurants, public offices, means of transportation) and outdoor when social distancing cannot be guaranteed, had an additional relevant impact on the pandemic control.

The proposed attribution of average risk classes for each employment sector is an indication to raise shared awareness on the current health emergency. Starting from the proposed approach each company will evaluate the specific risk and mitigate it through a tailored prevention strategy. Specificity and complexity of the single business areas should be taken into account, in particular for small and medium-sized enterprises.

The prevention system at national and corporate level, developed over time according to the EU OSH framework Directive 89/391/EEC and consolidated at national level through the Italian Legislative Decree 81/08, provides the natural framework to carry out an integrated approach in the evaluation and management of risks during the pandemic emergency. In line with the processes of risk evaluation and management regulated by law, general and specific measures must be adopted, commensurate with the risk of exposure to SARS-CoV-2 in work environments and favoring measures of primary prevention.

In the view of an integrated and participatory approach for the enforcement of the procedures identified, it is paramount the involvement of all the subjects responsible for health and safety at work–occupational health physicians (OHP), health and safety managers, workers' health and safety representatives. They must cooperate with the employer in monitoring and enforcing such measures.

A strong involvement of OHP and the organization of an "exceptional health surveillance" system during the COVID-19 health emergency and in the post-lockdown phase, need to be put in place, including careful measures to protect the health of fragile categories of workers (e.g. higher age groups, and of those individuals affected by one or more chronic-degenerative diseases).

OHPs will play a key role in all the activities related to risk assessment and health surveillance connected to SARS-CoV-2 infection. They will also be requested for an active contribution in the reintegration at work of those individuals with a history of SARS-CoV-2 infection. COVID-19 cases with pneumonia or severe acute respiratory syndrome may suffer a decreased lung capacity (even up to 20–30%) as a result of the illness, with possible need for prolonged respiratory physiotherapy [29]. Also prolonged hospitalization in Intensive Care Units and induced coma may be responsible for neurological disorders and behavioural changes, which need to be carefully considered and managed in the fit for work judgement [30].

Furthermore, the conscious and active participation of employees can bring effective results with beneficial effects also outside the working environment. It is also necessary to highlight that the perception of this risk, due to its exceptional nature and enormous impact, generates in the workers a feeling of insecurity that can also operate on other risks. Therefore, proper risk management and communication, alongside all other solutions adopted, can create a feeling of awareness and adequacy of the measures.

## Conclusions

Reactivation of businesses after the lockdown introduced several challenges for control of the pandemic, but at the same time presented an opportunity to extend the benefits of cost-effective measures to the community at large.

However, decisions on re-opening needed to follow a stepwise approach, including risk-based criteria to identify eligible sectors and allowing adequate intervals between phases to assess the impact of each one on control of the pandemic.

The prevention approach that has been proposed requires strong support from the national prevention system, in the offering of adequate information and training tools based on scientific evidence. It is also necessary to promote appropriate communication, even for risk perception, and actions are to be undertaken to contrast social stigma.

It is needed to further investigate the infection phenomenon and its impact on social and health sector to reinforce all those measures that are necessary to guarantee health protection of all workers. It will be necessary to consolidate remote working and to reinforce organizational support also with coaching and training tools. This will help in containing the risk of infection without compromising productivity, both for public administration and for service sector, due regard being given to the nature of production processes. Vulnerable workers' protection is an essential point also because of the peculiarity of the disease. Provision should be made to prevent exclusion of such workers from the world of work. Finally, epidemiological studies on seroprevalence, including elements related to occupational variables, will constitute further important contribution to the context analysis.

It is indispensable and fundamental that the entire proposed framework is coherently included in all epidemic containment policies, with particular reference to specific measures to prevent the emergence of new epidemic clusters.

In conclusion, the proposed approach will contribute to re-thinking how work is organized, also to include innovation, with integration in the OSH national system. The model will contribute to the prevention and the early identification of the outbreaks in the workplace in the future stages of the pandemic.

## Supporting information

**S1 Appendix.**
(DOCX)

## Author Contributions

**Conceptualization:** Sergio Iavicoli.

**Data curation:** Fabio Boccuni, Bruna Maria Rondinone.

**Formal analysis:** Fabio Boccuni, Giuliana Buresti, Diana Gagliardi, Benedetta Persechino, Antonio Valenti, Bruna Maria Rondinone.

**Methodology:** Sergio Iavicoli, Fabio Boccuni, Giuliana Buresti, Diana Gagliardi, Benedetta Persechino, Antonio Valenti, Bruna Maria Rondinone.

**Supervision:** Sergio Iavicoli, Bruna Maria Rondinone.

**Validation:** Sergio Iavicoli.

**Writing – original draft:** Sergio Iavicoli, Fabio Boccuni, Giuliana Buresti, Diana Gagliardi, Benedetta Persechino, Antonio Valenti, Bruna Maria Rondinone.

**Writing – review & editing:** Sergio Iavicoli, Fabio Boccuni, Giuliana Buresti, Diana Gagliardi, Benedetta Persechino, Antonio Valenti, Bruna Maria Rondinone.

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
