## [Decision Letter · Decision Letter 0]

8 Mar 2021

Risk assessment at work and prevention strategies on COVID-19 in Italy.

PONE-D-20-21804

Dear Dr. BOCCUNI,

We’re pleased to inform you that your manuscript has been judged scientifically suitable for publication and will be formally accepted for publication once it meets all outstanding technical requirements.

Kind regards,

Khin Thet Wai, MBBS, MPH, MA (Population & Family Planning Res.)

Academic Editor

PLOS ONE

Reviewers' comments:

Reviewer's Responses to Questions

**Comments to the Author**

1. Is the manuscript technically sound, and do the data support the conclusions?

Reviewer #1: Yes

Reviewer #2: Yes

2. Has the statistical analysis been performed appropriately and rigorously? 

Reviewer #1: N/A

Reviewer #2: N/A

3. Have the authors made all data underlying the findings in their manuscript fully available?

Reviewer #1: Yes

Reviewer #2: Yes

4. Is the manuscript presented in an intelligible fashion and written in standard English?

Reviewer #1: Yes

Reviewer #2: Yes

5. Review Comments to the Author

Reviewer #1: This paper deserve to be published since it provide important aspect of COVID-19 prevention and use of the risk assessment method which could also be replicated in similar as well as different settings with adaptation.

Reviewer #2: Article that deals with a fundamental topic in the period of the SARS-COV-2 pandemic, establishing parameters that allow the effective assessment of contagion within companies, allowing the development of anti-contagion procedures more suitable for various workplaces.

6. PLOS authors have the option to publish the peer review history of their article (what does this mean?). If published, this will include your full peer review and any attached files.

Reviewer #1: No

Reviewer #2: No

---

## [Editor Report · Acceptance letter]

10 Mar 2021

PONE-D-20-21804 

Risk assessment at work and prevention strategies on COVID-19 in Italy 

Dear Dr. Boccuni:

I'm pleased to inform you that your manuscript has been deemed suitable for publication in PLOS ONE. Congratulations! Your manuscript is now with our production department. 

Kind regards, 

on behalf of

Dr. Khin Thet Wai 

Academic Editor

PLOS ONE